# Genetic Characteristics and Pathogenicity of a Novel Porcine Epidemic Diarrhea Virus with a Naturally Occurring Truncated ORF3 Gene

**DOI:** 10.3390/v14030487

**Published:** 2022-02-27

**Authors:** Yuan-Hang Zhang, Hong-Xuan Li, Xi-Meng Chen, Liu-Hui Zhang, You-Yi Zhao, Ai-Fang Luo, Yu-Rong Yang, Lan-Lan Zheng, Hong-Ying Chen

**Affiliations:** 1Zhengzhou Major Pig Disease Prevention and Control Laboratory, College of Veterinary Medicine, Henan Agricultural University, Zhengdong New District Longzi Lake 15#, Zhengzhou 450046, China; zyhh105@163.com (Y.-H.Z.); lhxx1999@163.com (H.-X.L.); simonarchimonde@126.com (X.-M.C.); zlhgyw235@163.com (L.-H.Z.); zyy2412495001@126.com (Y.-Y.Z.); yangyu7712@sina.com (Y.-R.Y.); 2Henan Houyi Biological Engineering Co., Ltd., Zhengzhou Airport Economic Comprehensive Experimental Zone Jingang Avenue, Zhengzhou 451161, China; hcy211zjj@163.com

**Keywords:** porcine epidemic diarrhea virus, truncated ORF3, molecular characteristics, pathogenicity, virulence

## Abstract

Porcine epidemic diarrhea virus (PEDV) is the major pathogen that causes diarrhea and high mortality in newborn piglets, with devastating impact on the pig industry. To further understand the molecular epidemiology and genetic diversity of PEDV field strains, in this study the complete genomes of four PEDV variants (HN2021, CH-HNYY-2018, CH-SXWS-2018, and CH-HNKF-2016) obtained from immunized pig farms in central China between 2016 to 2021 were characterized and analyzed. Phylogenetic analysis of the genome and S gene showed that the four strains identified in the present study had evolved into the subgroup G2a, but were distant from the vaccine strain CV777. Additionally, it was noteworthy that a new PEDV strain (named HN2021) belonging to the G2a PEDV subgroup was successfully isolated in vitro and it was further confirmed by RT-PCR that this isolate had a large natural deletion at 207–373 nt of the ORF3 gene, which has never been reported before. Particularly, in terms of pathogenicity evaluation, colostrum deprivation piglets challenged with PEDV HN2021 showed severe diarrhea and high mortality, confirming that PEDV HN2021 was a virulent strain. Hence, PEDV strain HN2021 of subgroup G2a presents a promising vaccine candidate for the control of recurring porcine epidemic diarrhea (PED) in China. This study lays the foundation for better understanding of the genetic evolution and molecular pathogenesis of PEDV.

## 1. Introduction

Porcine epidemic diarrhea (PED) was originally reported in Belgium and England in the late 1970s [1]. Porcine epidemic diarrhea virus (PEDV), the etiological agent of PED, identified as a viral agent distinct from transmissible gastroenteritis virus (TGEV), is responsible for devastating enteric infection with significant morbidity and mortality in piglets [2]. At the end of 2010, a PEDV outbreak occurred in several pig-producing provinces in southern China where swine had been immunized with attenuated PEDV (CV777) [3,4]. Since then, the disease has spread throughout other provinces, which has resulted in enormous economic losses to the pork industry in China [5]. Despite its notorious reputation in Asia, PED was not well-recognized worldwide until the disease hit the United States in 2013, a variant PEDV called S-INDEL strain (S-INDEL standing for insertions/deletions in the S gene) rapidly propagating across 34 states of America and Canada [6,7,8]. Ever since, it has returned to devastate the swine industry in Asia despite the use of CV777-based vaccines. The emergence and re-emergence of PED have sustained a tremendous threat worldwide, and it is considered one of the most economically important diseases in countries with intensive swine industries [9,10].

PEDV is an enveloped, single-stranded RNA virus belonging to the order *Nidovirales*, family *Coronaviridae* and genus *Alphacoronavirus* [11]. The genome of PEDV is approximately 28 kb in size and includes seven open reading frames (ORFs) encoding three nonstructural proteins (replicase 1a,1b and ORF3) and four structural proteins (spike (S), envelope (E), membrane (M) and nucleoprotein(N)) [12]. Point mutations, gene insertions, and deletions of the PEDV genome have been frequently reported, indicating that genetic variation has occurred in the global PEDV pandemic strain, which led to its genetic diversity [13,14,15]. Based on the nucleotide sequences of the viral complete genomes, PEDV strains are classified into the classic G1 group and the variant G2 group. The G1 group includes the subgroups G1a and G1b, and the G2 group consists of the subgroups G2a and G2b [15,16]. The PEDV subgroups G2a and G2b circulating in China are highly pathogenic to suckling piglets and exhibit serum cross-neutralization activity or cross-immune protection. Moreover, it was found that the G2a subgroup strains of PEDV, compared with the G2b subgroup strains, could resist attack by homologous PEDV strains by mediating better immune protection [10,17]. 

To further investigate genetic diversity and infectivity of PEDV in central China, the whole genomes of four PEDV strains, HN2021, CH-HNYY-2018, CH-SXWS-2018 and CH-HNKF-2016, were obtained from diseased piglets during outbreaks of diarrhea between 2016 and 2021 on immunized farms in central China. Among them, a new PEDV strain (called HN2021) belonging to the G2a PEDV subgroup, with a novel naturally occurring truncated ORF3 gene, was successfully isolated and proved to be highly virulent to suckling piglets. The objective of the current study was to evaluate the genetic relationships, variant characteristics, and potential recombination features of the four PEDV strains, and elucidate the molecular characteristics and pathogenicity of the PEDV strain HN2021. Our findings could provide valuable information for PED outbreaks and shed light on its prevention and control in central China.

## 2. Materials and Methods

### 2.1. Virus, Clinical Samples Collection

The PEDV-HNCADC-2017 strain with intact ORF3 gene was kindly provided by Dr. Xue-li Zhao, Henan Provincial Animal Disease Prevention and Control Center, and served as the control virulent strain. Intestine and fecal samples were collected from diseased piglets during outbreaks of diarrhea on immunized farms in Henan, China from 2016 to 2021. Tissues were immersed in Dulbecco’s modified Eagle medium (DMEM). The sample homogenates were then frozen and thawed three times. After centrifugation, the supernatants were stored at −80 °C until use.

The research protocols for animal experiments were approved by the Animal Care and Use Committee of Henan Agricultural University (LLSC100085) and performed in accordance with the ‘Guidelines for Experimental Animals’ of the Ministry of Science and Technology (Beijing, China).

### 2.2. RT-PCR and Sequencing

The viral RNA of supernatants was extracted using TRIzol Reagent (Takara, Dalian, China) according to the manufacturer’s instructions. Viral cDNA was acquired via reverse transcription using HiScript II 1st Strand cDNA Synthesis Kit (Vazyme, Nanjing, China) following the manufacturer’s instructions. The cDNA was screened by RT-qPCR (quantitative real-time RT-PCR), the primers targeting the PEDV M gene were designed (Appendix A), and RT-qPCR was conducted by using the Premix Ex Taq™ kit (TaKaRa) on a real-time thermocycler (CFX96™ Optics Module; BIO-RAD; California, CA, USA). For ORF3 amplification, the primers against ORF3 gene (Appendix A) were used [18]. Then, a total of 20 pairs of primers were designed to amplify the complete genome of the PEDV strain by conventional RT-PCR (Appendix A), and the positive cDNA was selected and amplified for complete genome. After purification, the products were cloned into pMD-18T vector (TaKaRa, Dalian, China) for sequencing.

### 2.3. Phylogenetic and Recombination Analysis

Multiple nucleotide (nt) and amino acid (aa) sequence alignments of the entire genomes, S and ORF3 genes of the PEDV strains were assembled and analyzed by applying DNAStar and MEGA7 software, respectively. The corresponding phylogenetic trees were generated by the neighbor-joining method in MEGA7 software with a p-distance model, using 1000 bootstrap replicates. All trees were annotated and visualized by using the iTOL v.4 online tool (Interactive Tree of Life, http://itol.embl.de/, accessed on 15 August 2021). Protein structure was generated using SWISS-MODEL (https://swissmodel.expasy.org, accesses on 22 August 2021). The genome sequences of 37 PEDV reference strains deposited in the GenBank database were downloaded and are shown in Table 1. Potential recombination in the whole genomes of 41 PEDV strains in this study was assessed by the Recombination Detection Program v4 (RDP4).

### 2.4. Virus Isolation, Plaque Purification

Virus isolation of PEDV was attempted on Vero cells as previously described [19]. Briefly, the tissue supernatants were filtered through a 0.22-µm membrane, and homogenate supernatants and trypsin (10 mg/mL) were inoculated into monolayers of Vero cells, which were then incubated in a 37 °C incubator supplemented with 5% CO_2_. Until the obvious cytopathic effects (CPEs) were observed, cells were harvested, thawed, and refrozen multiple times. Then, the harvested virus supernatant was inoculated into the newly prepared Vero cells for subculture and propagated continuously for 10 generations. 

For plaque purification of the PEDV HN2021 strain, monolayers of Vero cells were inoculated with this strain. After incubation for 2 h, the cells were overlaid with 1.5% agarose. Plaques were stained with neutral red dye at 48 h post-inoculation (hpi).

### 2.5. Immunofluorescence Assay and Viral Growth Curve

The tenth generation of plaque purification virus was selected for subsequent experiments. The cells in 6-well plates were fixed with cold anhydrous ethanol when 70% of cells showed CPE. Next, an immunofluorescence assay (IFA) was performed with a rabbit anti-S protein polyclonal antibody (prepared in our lab, diluted 1:1000) of PEDV for 1 h at 37 °C followed by a fluorescein isothiocyanate-conjugated goat anti-rabbit antibody against immunoglobulin G (Proteintech, Rosemont, IL, USA; dilution: 1:1000). The plates were examined using a fluorescence microscope.

The growth curve of HN2021 was determined in Vero cells according to the median tissue culture infection dose (TCID_50_). In short, Vero cells were seeded in a 96-well plate in 100 μL of medium at a density of 10^5^ cells per well, and incubated for 48 h under 5% CO_2_ at 37 °C. Later, the medium was removed and 100 μL of 10-fold serially diluted virus harvested at different time points was added to each well. The CPE was checked every 12 h for 3–5 days after inoculation. The virus titer was determined according to the method of Reed and Muench [20].

### 2.6. Pig Infection Experiments

To determine the virulence of PEDV isolate, nine three-day-old piglets, which were confirmed negative for PEDV, TGEV, porcine delta coronavirus, and porcine rotaviruses by virus-specific RT-PCRs of rectal swabs, were chosen for the challenge experiment. During the acclimation period, all piglets were active with normal fecal consistency, and showed no clinical symptoms. The PEDV-HNCADC-2017 strain with intact ORF3 gene served as the control virulent strain. Pigs were randomly assigned to three groups (*n* = 3): the HN2021-inoculated group, PEDV-HNCADC-2017-inoculated group, and DMEM-inoculated control group. All the piglets were reared in separate rooms and provided with sterilized feed and water. Following a 2-day acclimation period, piglets (5 days old) in the virus-inoculated groups received a 2 mL dose of 10^5.0^ TCID_50_/mL of each of the viruses orally. The sham-inoculated pigs were administered cell culture media as a placebo. 

Body temperatures and weights of piglets were recorded before inoculation (−12 hpi) and then every 24 h until death. Animals were monitored daily for clinical signs of vomiting, diarrhea, and mortality throughout the experiment. Stool samples from pigs in all groups were collected daily with swabs prior to and after inoculation. All the collected fecal samples were diluted 5-fold with DMEM and centrifuged at 4200× *g* at 4 °C for 10 min. The supernatants were collected for viral RNA extraction and then subjected to real-time RT-PCR to detect the presence of PEDV shedding. A clinical significance score was determined with the following scoring criteria based on visual examination for 7 days post-inoculation (dpi) used to measure diarrheal severity: 0, normal and no diarrhea; 1, mild and fluidic feces; 2, moderate watery diarrhea; 3, severe watery and projectile diarrhea; 4, death. 

Piglets were necropsied upon death after challenge throughout the study, whereas all surviving pigs from the challenge and control groups were euthanized at 7 dpi for post-mortem examinations. Fresh and formalin-fixed samples, including duodenum, jejunum and ileum, were collected during the necropsy. The fresh samples were used for viral distribution detection, and the formalin-fixed tissue samples were used for the pathological examination, including histopathological sections and immunohistochemistry (IHC). 

## 3. Results

### 3.1. Complete Genomic Characterization

The complete genome sequences of four PEDV strains (HN2021, CH-HNYY-2018, CH-SXWS-2018, and CH-HNKF-2016) were derived by combination of 20 overlapping cDNA fragments.

Sequence comparisons of the whole genomes showed that nucleotide homologies between the four strains examined in this study and PEDV reference strains ranged from 95.8–99.2%, 96.3–99.0%, 96.3–99.2%, 96.3–99.0%, respectively. These four strains shared a general low similarity (95.6–97.5%) with the reference strains belonging to the G1 group (including the CV777, variant DR13, LZC, etc.) and 97.1–99.2% identity with the G2b reference strains. On the contrary, they were highly homologous (97.8–99.2%) to those strains in group G2a which consisted of most Chinese isolates, and had 97.9–98.6% identity with each other (Table 1). Furthermore, HN2021 shared the highest homology (99.2%) with CO-P14-IC clustered in G2a, which was a cell-adapted strain isolated in the USA with a truncated ORF3 gene. 

In this study, the S genes of the four strains were 4161bp in length, encoding a 1386-aa protein. A multiple nt sequence alignment illustrated that HN2021 exhibited 95.2–98.8% identity with the G2a strains, 96.8–97.5% identity with the G2b strains, and 91.9–93.6% identity with the G1 strains. However, HN2021 possessed 97.7%, 97.5%, and 97.4% identity with CH-HNYY-2018, CH-SXWS-2018, CH-HNKF-2016, respectively (Table 1). Notably, when compared with the S protein of CV777, the four strains displayed common aa deletions (^163^DI^164^) or insertions (^59^QGVN^62^ and^139^N^140^) frequently occurring in the N-terminal domain (NTD, 19-233aa) of the S protein, among which a distinct change (N140Y) in the insertion site was observed in CH-HNYY-2018 (Figure 1a). It has been demonstrated that four major neutralizing epitopes (COE(499–638 aa), SS2(748–755 aa), SS6(764–771 aa) and 2C10(1368–1374 aa)) are present on the surface of the PEDV S protein [21]. Interestingly, some aa mutations were located within the neutralizing epitopes of PEDV (Table 2). Additionally, these deletions, insertions and mutations directly led to a structural change at some parts of the HN2021 S protein compared to the CV777 S protein (Figure 2a,b).

When it comes to the ORF3 genes, the length of three strains is 675 bp encoding 224 aa, except that the HN2021 is 508 bp in length encoding 68 aa. Subsequently, a sketch map of the ORF3 comparisons of the four strains together with CV777, attenuated DR13, CO/P14/IC and the HN2021 is shown in Figure 1b. The results revealed that the naturally truncated strain HN2021 exhibited 167 continuous nucleotide deletions at 207–373 nt. Then, the ORF3 aa identity of the four strains in this study was compared with 37 reference strains, illustrating that HN2021 was 94.2–98.6% identical to other PEDV strains, and the remaining three strains in this study exhibited 91.3–99.6% aa similarity with the reference strains. After alignment, it was shown that HN2021 owned the shortest ORF3 protein among all strains in this study, which was 68 aa, with 156 aa deletions at C’ end (Figure 1c). Furthermore, in addition to shared mutations, one unique substitution (R75H) was found in the ORF3 protein of the CH-HNKF-2016 strain that was absent in other strains.

### 3.2. Phylogenetic and Recombination Analysis

On the basis of the genomes for the S and ORF3 genes of PEDV strains, phylogenetic trees were constructed. The genome-based phylogenetic tree (Figure 3a) showed that the 41 PEDV strains were classified into two groups: G1 (classical strains) and G2 (variant strains). G1 contained nine isolates: CV777, SD-M, attenuated or virulent DR13, JS2008, KC189944, HIJBY, LZC, and PEDV-SX from China. Group 2 was composed of two subgroups G2a, G2b. The isolate HN2021 and the other three strains identified in the present study had evolved into the subgroup G2a, which consisted of most PEDV reference strains from China between 2011–2018. PEDV strains of the subgroup G2a formed different clades, exhibiting genetic diversities. The phylogenetic tree for the S gene (Figure 3b) showed a similar grouping structure as the tree generated from the PEDV entire genomes. The strains CH-HNYY-2018, CH-SXWS-2018 and CH-HNKF-2016 in this study formed an independent clade within the same subgroup, which clustered closely around the Chinese isolates such as the HNAY2016 strain, but were distant from the epidemic strains in the USA. In contrast, the strain HN2021 had a near relationship to the isolates in America, such as CO-P14-IC and USA-Colorado-2013. Phylogenetic analysis of the ORF3 gene (Figure 3c) indicated that PEDV strains were clustered into three groups. The major PEDV strains with truncated ORF3 gene, such as attenuated DR13, HJBY, JS2008 and PEDV-SX were grouped in G1 together with CV777 and virulent DR13. The G2 included two subgroups G2a and G2b. The CH-HNYY-2018, CH-SXWS-2018, CH-HNKF-2016 together with most Chinese strains and some foreign strains such as PC22A, USA/Colorado/2013 and KNU-1807 were clustered into G2a, which covered most strains with an intact ORF3 gene. Intriguingly, truncated ORF3 gene strains HN2021 and CO-P14-IC were also clustered into the G2a subgroup. Generally, genomic phylogenetic analysis revealed that the novel PEDV strain HN2021 exhibited divergence from the reference strains of group G1 containing the classical vaccine strain CV777, but showed high similarity with reference variant strains from the subgroups G2a. 

A whole genome recombination analysis between the four strains and 37 reference strains was performed using RDP4 software. In general, a putative recombination event was found in CH-HNYY-2018. As shown in Figure 3d, CH-HNYY-2018might arise by the recombination of CH_hubei_2016 and GDS28 strains, which was supported by six programs. Furthermore, the identified putative breakpoints were located in the ORF1b and S region (17,937–21,538 nt). 

### 3.3. Viral Isolation and Identification

Among the PEDV-positive samples, a sample (HN2021) from Xuchang city, Henan province, appeared as a remarkable CPE which was characterized by the rounding up, enlarging and detachment of cells as well as the formation of syncytia on Vero cells at the beginning of passage 3 (P3) compared with control cells (Figure 4a,b). It was further confirmed by RT-PCR targeting as the ORF3 gene and named HN2021 (Figure 4c). When serially passaged for 10 generations in Vero cells, the isolate HN2021 (P10) was successfully purified by plaque cloning for follow-up experiment (Figure 4d). The IFA result based on polyclonal antibody against PEDV S protein was observed in the inoculated cells, but not in the non-infected cells (Figure 4e). After verification by RT-PCR and IFA, the viral growth curve was generated based on the TCID_50_ values at 12, 24, 36, 48, 60 hpi, respectively. The results showed that the titer of strain HN2021 (P10) reached the maximum (10^4.6^ TCID_50_/mL) at 48 hpi (Figure 4f).

### 3.4. Pathogenicity of PEDV Isolates in New-Born Piglets

Since high-passage derivatives of PEDV experimentally exhibit reduced pathogenicity in pigs, and a large deletion in the ORF3 region is present in attenuated or live vaccine strains, the deletion pattern identified in the present study is most likely to be associated with the virulence of HN2021. Thus, the HN2021 strain was chosen for subsequent animal studies to assess the pathogenicity in vivo. Following challenge, none of the negative-control piglets developed clinical signs typical of PEDV throughout the pathogenicity study. Nevertheless, HN2021-challenged (group 1) and PEDV-HNCADC-2017-challenged piglets (group 2) exhibited clinical signs including lethargy and diarrheic feces by 1 dpi, and experienced severe watery diarrhea with vomiting thereafter. Notably, the fecal scores of group 1 showed no significant difference to those of group 2 throughout most of the duration of the experiment (Figure 5a).

PEDV-associated mortality occurred in inoculated animals in group 1 and 2 at 5 or 6 dpi, and the mortalities reached 33% (1/3) and 100% (3/3), respectively (Figure 5b). Necropsy examinations were performed immediately after the death of the infected piglets. Furthermore, viral RNA in fecal samples tested positive for PEDV in challenged groups (4–10 log10 genomic equivalents/mL), and both reached a peak at 5 dpi (Figure 5c). The viral load in the fecal from group 1 was higher than that of group 2, but there was no obvious significance. The viral loads in different segments of the small intestine including duodenum, jejunum, and ileum were determined. As shown in Figure 5d, the segments of duodenum, jejunum and ileum exhibited high viral loads in both group 1 and 2, ranging from 9–12 log10 genomic equivalents/mL, in which the viral loads of ileums were higher than those of the duodenums and jejunums, and this was not significant. Additionally, similar to fecal, the viral load of the small intestine in group 1 exceeded that of the group 2 insignificantly. Neither virus shedding in feces nor the viral load in tissues of the control group were detected.

All the infected piglets displayed typical lesions of PED and weight loss (Figure 5e). The wall of the small intestine was thin and transparent, distended and filled with curdled and undigested milk. By contrast, the intestinal organs of control piglets appeared grossly normal (Figure 6a–c). Microscopic examination of the small intestine in the two PEDV isolates infected piglets revealed the shortening, atrophy or the gross shedding of intestinal villi (Figure 6d–f). However, the pigs of the negative control group exhibited normal intestinal histo-pathologies. Furthermore, IHC staining revealed that PEDV antigen was predominant in the cytoplasm of epithelial cells in atrophied villi in all segments of the small intestines, but none of piglets in the control group was IHC positive (Figure 6g–i). Altogether, analogous to the PEDV-HNCADC-2017 strain, our data indicates that the HN2021 strain, which had a natural large deletion at the 207–373 nt position of the ORF3 gene, has been verified as a virulent strain causing severe diarrhea and high mortality in experimentally suckling piglets.

## 4. Discussion

Since the sudden emergence of PEDV in the US, this viral agent has become globally renowned and is now considered as an emerging or reemerging pathogen with economic effects on the world pork business [8]. Recently in China, despite substantial efforts, including a nationwide vaccination campaign for PEDV control, this devastating virus has menaced the domestic swine industry over the last decade and become endemic, thereby causing continuous financial disbenefits [10]. To prevent emerging and re-emerging PEDV-associated epizootics and enzootics, it is crucial not only to continue monitoring and characterizing the virus circulating in pig-producing regions but also to pursue unremitting exertion in developing more effective vaccines.

In the present study, the complete genome sequences of 4 PEDV strains were obtained from PED-affected pigs in disparate swine farms of Henan province. Among them, a novel PEDV strain (HN2021) containing naturally occurring truncated ORF3 gene with continuous deletions from 207–373 bp that encoded a 68 aa truncated ORF3 protein was isolated successfully in Vero cells. The genetic analysis of the genome showed that the PEDV strains fell into two genogroups, designated G1 (classical) and G2 (variant). G2 group was further subdivided into G2a and G2b subgroups in this study. Notably, the four strains determined in this study fell into genogroup G2a which consisted of the most recent Chinese isolates, and similar results were observed in analyses based on S and ORF3 genes. In addition, a recombination event between different PEDV subtypes was identified in the genome of CH-HNYY-2018 at the ORF1b and S region (17,937–21,538 nt) This indicated that CH-HNYY-2018 might be derived from natural recombination between two Chinese PEDV mutants of different subtypes. Indeed, previous studies have provided evidence that recombination among different strains is a major evolutionary factor that could allow the emergence of new strains with altered virulence and immunogenicity [22,23]. The events determined in this study revealed that recombination between strains of different subtypes had presented itself in Henan province, which might increase the difficulty of prevention and control of PEDV.

When compared with the S protein of CV777, except for the common aa deletions or insertions, a distinct change (N140Y) at the insertion site located in the N-terminal domain (NTD, 19–233 aa) was observed in CH-HNYY-2018. In comparation with classical and vaccine strains, a series of aa insertions and mutations in the S gene may play an important role in the pathogenicity and antigenicity of the new PEDV variants [10]. The antigenicity analysis of the S gene showed that the four neutralizing epitopes (COE, SS2, SS6, and 2C10) share unique genetic features compared with the CV777 strain [21,24]. Some aa mutations were also widely located within the neutralizing epitopes of PEDV, which directly led to a structural change at some parts of the HN2021 S protein compared to the CV777 S protein. Similar to other coronavirus S proteins, the PEDV S protein plays a critical role in viral entry and production of neutralization antibodies in the host [25,26,27]. This transformation might be associated with its frequent interaction with host cells. Whether these aa substitutions and the N-glycosylation site substitutions influence the antigenicity and pathogenicity of PEDV remains to be investigated.

While coronaviruses generally express various accessory proteins from the 3′ part of their genomes, the single accessory ORF occurring in PEDV is the only one conserved in all coronaviruses [28,29]. PEDV ORF3 encodes an anion channel protein of 224 aa with a theoretical molecular weight of 25 kDa, predicted to be a multi-spanning membrane protein [30]. This protein is prone to undergoing deletion or mutation when the virus is adapted to grow in cell culture, e.g., by serial passaging. Additionally, the classic vaccine strains (attenuated CV777, attenuated DR13) had 49 nucleotide deletions at 245–293 nt, giving rise to a naturally truncated ORF3 protein of 91 residues, and have been proposed to be responsible for the reduced pathogenicity of these viruses. Field isolates with large deletions of ORF3 have also been documented [31]. However, naturally occurring truncated ORF3 gene of PEDV strains has been rarely reported in central China. Through the comparative analysis of ORF3 genes, it was noteworthy that the HN2021 isolate had 167 continuous nucleotide deletions from 207 to 373 nt. With such a frame-shifting deletion, the translation of HN2021 ORF3 protein is predicted to terminate much earlier and be 68 aa in length. More strikingly, it was first discovered and isolated successfully in central China. Previously, JS2008 was considered to be a recombination of a vaccine strain and a PEDV variant strain [32]. It is not possible to speculate whether the HN2021 strain with naturally truncated ORF3 genes is a recombination of the vaccine strain and a PEDV variant strain, and this needs further confirmation. As aforementioned, several attenuated vaccine strains retain large deletions in ORF3 and encode truncated ORF3 variants, suggesting that this genotypic feature is associated with the loss of PEDV virulence.

As there are only a few reports on the characteristics of the ORF3 protein, little is actually known about its precise functioning during PEDV infection. It has been demonstrated, by using reverse genetics systems, that the ORF3 is dispensable for viral growth in vitro [33,34]. Moreover, previous studies have demonstrated that the naturally occurring aa deletion could abolish the suppressive effect of some ORF3 variants on virus replication in vitro [35]. The naturally truncated ORF3 protein displayed a more apparent association with the S protein, and it may work in concert to regulate PEDV replication in vivo [36]. Generally, the naturally truncated form of ORF3 might cause the attenuation of PEDV in a natural host, but not all ORF3 truncations have been associated with reduced pathogenicity [37]. In view of the inconsistent literature regarding the role of ORF3 and its importance for viral pathogenicity, the HN2021 virus at selected passages was inoculated into piglets to evaluate its virulence/attenuation phenotypes so that the genetic changes potentially associated with virus attenuation could be identified.

In this study, the infection piglets in HN2021 and PEDV-HNCADC-2017 groups both showed severe symptoms, mainly characterized by watery diarrhoea and acute death. Most attenuated strains, which exhibited 49 nucleotide deletions at 245–293 nt of ORF3 gene and showed low pathogenicity in piglets, for instance, the attenuated CV777 and attenuated DR13, belonged to G1 genotype. Nevertheless, the truncated ORF3 HN2021 strain belonging to G2 genotype appeared to be highly virulent to suckling piglets. Similar results emerged in the 17GXCZ-1ORF3d strain belonging to G2 genotype, which has a naturally truncated ORF3 gene containing a continuous 382 nucleotide deletions from 172–554 nt and has been verified as a virulent strain causing severe diarrhea and high mortality in suckling piglets [18]. The genotype of PEDV is closely related to pathogenicity and immune protection [38]. Hence, we speculate that the genes affecting virus virulence may be changing with the variation in the S gene. More interestingly, a recent study revealed that the virulence of both KNU-141112 S DEL2/ORF3 and KNU-141112 S DEL5/ORF3 viruses belonging to G2b subtype with a large 46-nt deletion in the intergenic portion of S and ORF3 was remarkably diminished, indicating viral attenuation in the natural host, while distinct deletion mutants were obtained through cell culture adaptation [39,40]. However, compared with HN2021, KNU-141112 S DEL2/ORF3 and KNU-141112 S DEL5/ORF3 not only possessed incomplete ORF3 protein, but lack several amino acid residues at the S protein terminal, which might play a key role in viral attenuation, but this needs to be confirmed in further studies. Taken together, it is still plausible that other genetic mutations are involved in the pathogenicity of PEDV, implying the multigenic character of PEDV virulence. Future work should aim to address the questions of whether genetic drifts in other genes, especially in S, influences PEDV virulence, and what the specific role of the ORF3 protein is in viral replication and pathogenesis.

In summary, a novel PEDV strain (called HN2021) belonging to the G2a PEDV subgroup, with a novel naturally occurring truncated ORF3 gene was successfully isolated in vitro, and confirmed to be highly virulent to suckling piglets. Therefore, further research is needed to determine the impact of this deletion of ORF3 gene on the pathogenicity of the PEDV strain. In addition, the distribution of PEDV with a naturally truncated ORF3 gene in China or more globally needs further investigation, which will significantly contribute to the prevention and control of reemerging PED outbreaks worldwide.

## Figures and Tables

**Figure 1 viruses-14-00487-f001:**
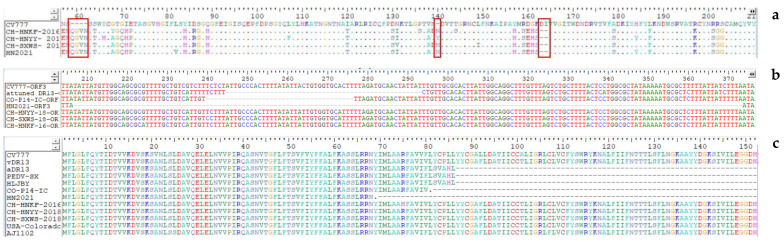
Partial genome comparison between the 4 PEDV strains in this study and the reference strains. (**a**) Compared with CV777 strain, the 4 strains displayed common 3 aa deletions or 5 aa insertions occurring in the N-terminal domain of the S protein, of which a distinct change (N140Y) in the insertion site was observed in CH-HNYY-2018. (**b**) The ORF3 gene of classic virulent strain CV777 was complete, and the attenuated DR13 strain had 49 nucleotide deletions at 245–293 nt, CO/P14/IC possessed 37 nucleotide deletions at 240–276 nt, the naturally truncated strain HN2021 exhibited 167 continuous nucleotide deletions at 207–373 nt. (**c**) Attenuated-DR13, HLJBY and PEDV-SX ORF3 proteins were 91 aa, which had 133 aa deletions at C’. CO/P14/IC ORF3 protein was 80 aa, with 144 aa deletions at C’ end, and the HN2021 owned the shortest ORF3 protein of all strains in this study, which was 68 aa, with 156 aa deletions at C’ end.

**Figure 2 viruses-14-00487-f002:**
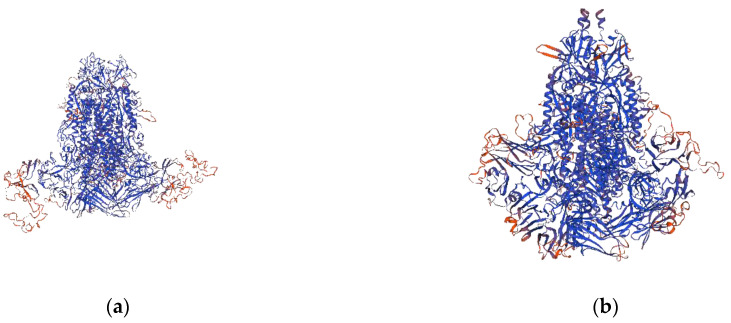
3D structure prediction of PEDV S protein. (**a**) modelling the 3D structure of the S protein of CV777; (**b**) modelling the 3D structure of the S protein of HN2021.

**Figure 3 viruses-14-00487-f003:**
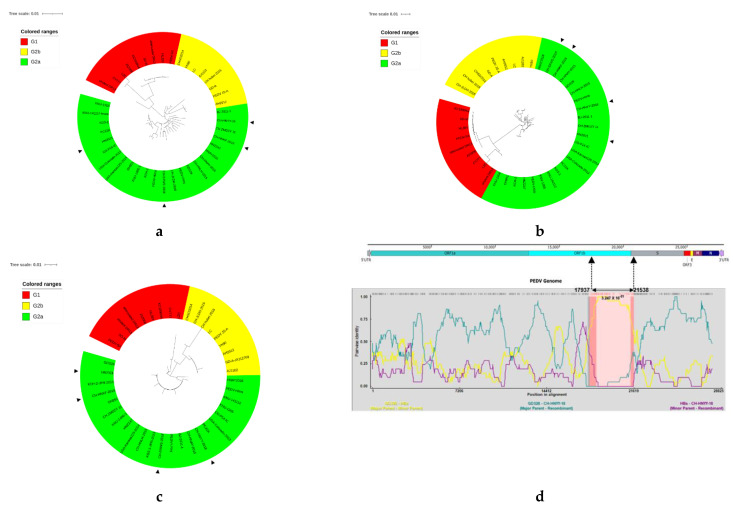
Genotyping and recombination analysis of the 41 PEDV strains in the present study based on different genes. Phylogenetic trees were constructed based on the aligned full-length genomic sequences (**a**), S gene (**b**) and ORF3 gene (**c**) by using the neighbor-joining method from MEGA 7.0, with 1000 bootstrap replicates. The trees were drawn to scale, with branch lengths measured in the number of substitutions per site. Group 1, G2a subgroup and G2b subgroup are colored in red, green and yellow, respectively. The triangle symbols represent the strains obtained in this study. (**d**) Recombination analysis of CH-HNYY-2018 with indicated PEDV strains. The result was described using the RDP method which was supported by >6 programs to further characterize the potential recombination events. The black arrow indicates the regions where a recombination event may occur.

**Figure 4 viruses-14-00487-f004:**
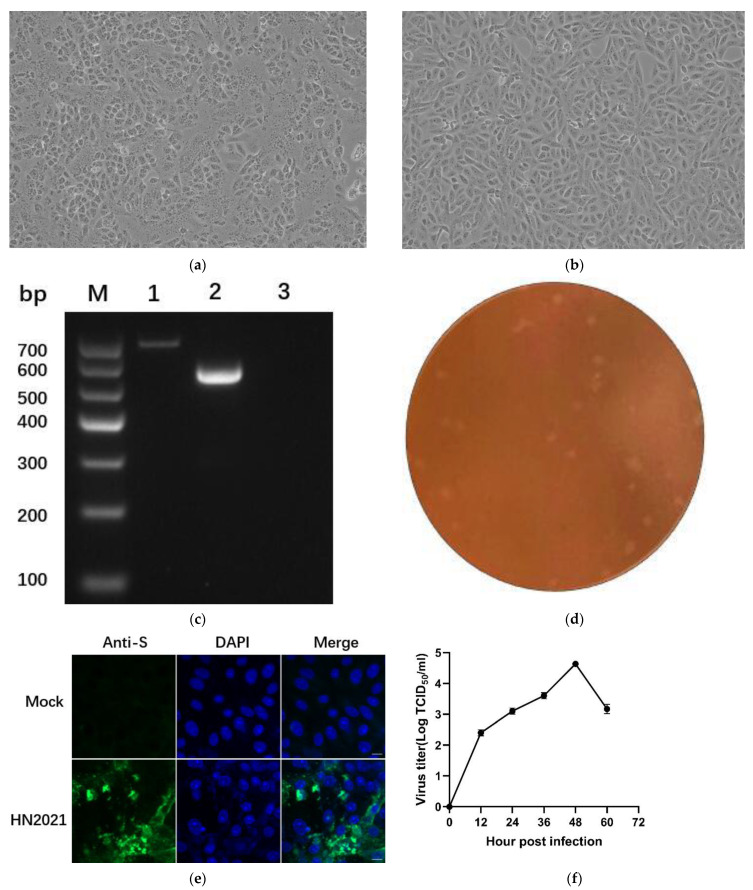
Virus isolation and purification of the HN2021 PEDV strains. CPE formation in Vero cells infected with (**a**) the HN2021 showing as rounded and clustered at 48 hpi (200×). (**b**) Vero cells control (200×). (**c**) Detection and amplification of the ORF3 gene in PEDV strains. M: DL Marker I; Lane 1: PEDV-HNCADC-2017 with intact ORF3 gene (~740 bp); Lane 2: HN2021 with a large genomic deletion (~508 bp); Lane 3: negative control. (**d**). Plaque purification of PEDV HN2021 strain. Monolayers of Vero cells were inoculated with PEDV HN2021 strain. After incubation for 2 h, the cells were overlaid with 1.5% agarose. Plaques were stained with neutral red dye at 48 hpi. (**e**) Detection of PEDV infection in Vero cells by IFA. (**f**) Growth curves of PEDV HN2021 isolate at passage 10 in Vero cells. Data is presented as mean ± SD by triplicates.

**Figure 5 viruses-14-00487-f005:**
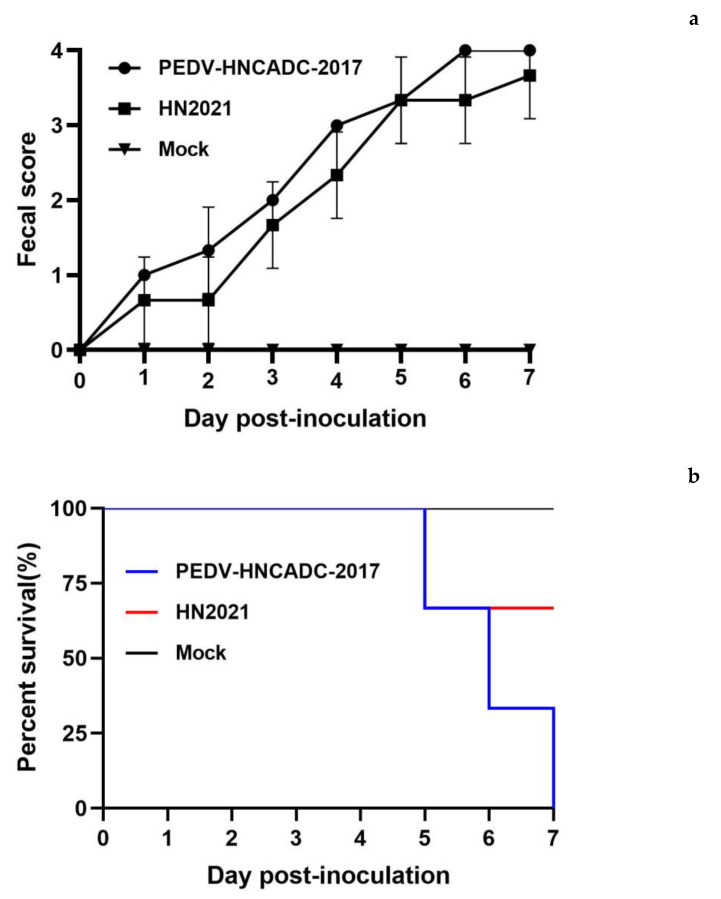
Pathogenicity analysis of PEDV isolates in new-born piglets. (**a**) The severity of diarrhea was scored based on clinical examination; 0, normal and no diarrhea; 1, mild and fluidic feces; 2, moderate watery diarrhea; 3, severe watery and projectile diarrhea; 4, death. (**b**) Survival rate of piglets in each group. (**c**) Fecal virus shedding in PEDV-challenged and control groups by RT-qPCR. (**d**) Viral RNA shedding titers of each group were determined by RT-qPCR in the duodenum, jejunum and ileum of piglets. (**e**) The piglet body weights between the two PEDV-challenged groups and control groups were compared. Error bars indicate the standard deviations from each group (*n* = 3). * *p* < 0.05.

**Figure 6 viruses-14-00487-f006:**
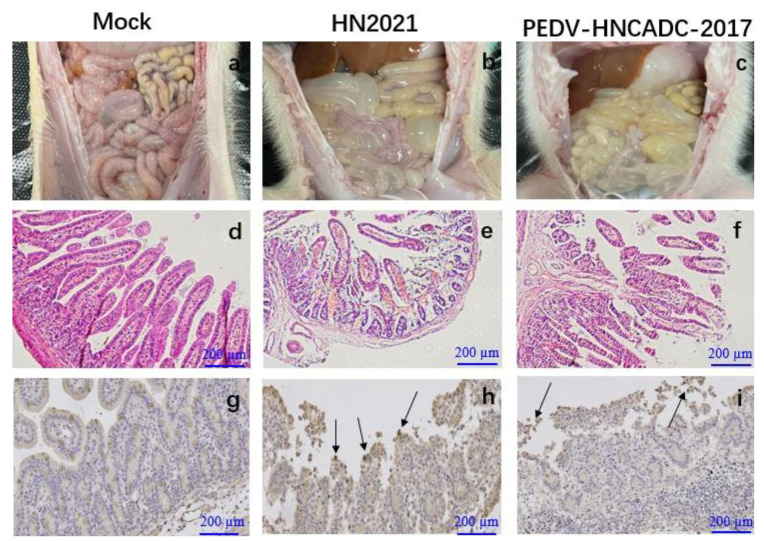
Lesions and IHC of small intestinal tissue sections from piglets inoculated with two types of PEDV. (**a**–**c**) Macroscopic lesions of the small intestines from challenged and control piglets. (**d**–**f**) Microscopic damage in Jejunum in piglets (Original magnifications: ×20). (**g**–**i**) PEDV antigen detection in different intestinal tissues of piglets infected with PEDV isolates by IHC (Original magnifications: ×20). Antigens are depicted as arrows. Scale bars are shown in each picture.

**Table 1 viruses-14-00487-t001:** Comparation of nucleotide and amino acid sequence identity (%) of different regions when comparing the 4 porcine epidemic diarrhea virus (PEDV) strains in this study and the representative strain.

Strain	Accession Number	Location	Time	HN2021 Identity			CH-HNYY-18 Identity			CH-SXWS-18 Identity			CH-HNKF-16 Identity		
				Genome	ORF3	S	Genome	ORF3	S	Genome	ORF3	S	Genome	ORF3	S
CV777	AF353511	Belgium	1978	96.0	95.7	92.8	96.6	96.0	93.2	96.7	96.0	93.5	96.7	95.6	93.3
virulent DR13	JQ023161	Korea	1999	96.9	98.6	93.6	97.4	99.1	94.1	97.5	99.1	94.4	97.5	98.7	94.1
attenuated DR13	JQ023162	Korea	2003	96.4	98.6	92.4	96.7	92.4	92.8	96.7	92.4	93.0	96.7	91.3	93.0
KNU-1305	KJ662670	Korea	2013	99.0	98.6	98.6	98.8	99.1	98.7	98.9	99.1	98.6	98.8	98.7	98.5
USA-Colorado-2013	KF272920	USA	2013	99.1	98.6	98.8	98.9	99.1	98.8	99.0	99.1	98.8	98.6	98.7	98.6
OH851	KJ399978	USA	2013	98.3	98.6	95.2	98.7	99.6	95.6	98.5	99.6	95.9	98.6	99.1	95.7
PC22A	KY499262	USA	2013	98.9	98.6	98.8	98.8	99.6	98.8	98.9	99.6	98.7	98.8	99.1	98.6
KNU-1406-1	KM403155	Korea	2014	98.3	98.6	95.2	98.6	99.6	95.5	98.4	99.6	95.9	98.3	99.1	95.6
USA-Kansas125-2014	KJ645701	USA	2014	99.1	98.6	98.4	98.7	99.6	98.6	98.8	99.6	98.4	98.0	99.1	98.3
CO-P14-IC	KU558702	USA	2014	99.2	98.6	98.8	98.6	98.8	98.1	98.7	98.8	98.1	98.5	97.5	97.9
KCH-2	LC063845	Japan	2014	98.1	98.6	95.3	98.0	99.6	95.7	98.1	99.6	96.0	98.4	99.1	95.7
KGS-1	LC063814	Japan	2014	98.7	98.6	98.6	98.4	99.6	98.7	98.5	99.6	98.6	98.0	99.1	98.5
LZC	EF185992	China	2006	95.8	94.2	91.9	96.3	94.2	92.3	96.4	94.2	92.5	96.5	93.8	92.4
JS2008	KC109141	China	2008	96.6	98.6	92.6	96.8	92.4	93.0	96.9	92.4	93.6	96.9	91.3	93.2
BJ-2011-1	JN825712	China	2011	98.5	98.6	98.4	98.8	99.1	98.8	98.9	99.1	98.7	98.8	98.7	98.6
AJ1102	JX188454	China	2011	97.9	97.1	97.5	98.4	96.9	98.1	98.5	96.9	98.1	98.8	96.4	97.8
CH-ZMZY-11	KC196276	China	2011	98.2	97.1	97.8	99.0	99.1	98.2	98.9	99.1	98.1	98.7	98.7	98.0
SD-M	JX560761	China	2012	96.5	98.6	92.1	96.7	92.4	92.5	96.8	92.4	93.0	96.8	91.3	92.6
LC	JX489155	China	2012	97.9	97.1	97.5	98.3	96.9	98.1	98.5	96.9	98.1	98.3	96.4	97.8
KC189944	KC189944	China	2012	96.3	98.6	92.0	96.5	92.4	92.4	96.6	92.4	92.9	96.5	91.3	92.5
GD-A GDS28	JX112709 MH726372	China China	2012 2012	97.5 98.3	97.1 98.6	97.5 97.5	98.0 98.7	96.0 99.6	98.0 98.1	98.1 99.2	96.0 99.6	97.8 98.6	97.9 99.0	95.6 99.1	97.5 98.5
CHSD2014	KX791060	China	2014	98.2	97.1	97.1	98.4	98.2	97.6	98.6	98.2	97.7	96.3	97.8	97.4
AH2012	KU646831	China	2014	98.7	97.1	97.2	98.8	96.0	97.7	98.9	96.0	97.5	98.4	95.6	97.3
HLJBY	KP403802	China	2015	96.4	97.1	92.0	96.6	91.3	92.5	96.6	91.3	93.0	96.7	90.2	92.6
CH-HNAY-2015	KR809885	China	2015	98.1	98.6	97.8	98.8	99.1	98.1	99.0	99.1	98.8	98.9	98.7	98.8
YN90	KT021231	China	2015	98.1	97.1	97.2	98.5	91.7	97.7	98.6	91.7	97.6	98.6	91.0	97.4
CH-HNLH-2015	KT199103	China	2015	98.2	98.6	97.8	98.6	99.1	98.3	99.1	99.1	98.9	98.9	98.7	98.8
PEDV-LNsy	KY007140	China	2015	98.2	98.6	97.6	98.7	99.6	98.0	98.1	99.6	98.2	98.7	99.1	97.9
CH-JLDH-2016	MF346935	China	2016	98.0	97.1	97.5	98.5	96.9	98.0	99.2	96.9	98.6	98.6	96.4	98.6
CH_hubei_2016	KY928065	China	2016	97.1	94.2	96.8	97.7	96.0	97.5	98.1	96.0	98.1	98.3	95.6	98.0
HNZZ47	KX981440	China	2016	97.8	98.6	97.0	98.5	99.6	97.8	98.5	99.6	97.9	98.5	99.1	97.6
HNAY2016	MT338518	China	2016	98.0	97.1	97.8	98.6	98.7	98.4	98.9	98.7	98.8	98.8	98.2	98.6
PEDV-Hims	KY007139	China	2016	98.1	98.6	97.8	98.6	99.6	98.4	98.6	99.6	98.1	98.8	99.1	97.8
PEDV-SX	KY420075	China	2017	96.4	98.6	92.2	96.6	92.4	92.6	96.7	92.4	93.1	96.7	91.3	92.8
PEDV JS-A	MH748550	China	2017	98.1	97.1	97.2	97.9	96.4	97.6	98.0	96.4	97.5	97.9	96.0	97.2
HN2021	-	China	2021	100	100	100	98.1	98.6	97.7	98.1	98.6	97.5	98.0	98.6	97.4
CH-HNKF-2016	KY649107	China	2016	97.9	100	97.4	98.6	99.1	97.9	98.8	99.1	98.6	100	100	100
CH-SXWS-2018	MT090146	China	2018	98.1	98.6	97.5	98.6	99.6	98.3	100	100	100	98.8	99.1	98.6
CH-HNYY-2018	MT090145	China	2018	98.1	98.6	97.7	100	100	100	98.7	99.6	98.3	100	99.1	97.9

**Table 2 viruses-14-00487-t002:** Amino acid mutations in neutralizing epitopes of the putative S protein of 4 PEDV strains from this study compared with CV777.

Strain	Position of Amino Acid Point Mutation in Major Neutralizing Epitopes									
	522	526	528	532	554	599	610	617	640	771
CV777	A	L	S	V	T	G	A	L	I	I
HN2021	S	H	G	I	S	S	E	F	V	F
CH-HNYY-18	S	H	G	I	S	S	E	F	V	F
CH-SXWS-18	S	H	G	I	S	S	E	F	V	F
CH-HNKE-16	S	H	G	I	S	S	E	F	V	F

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
