# Peer review of "Genetic Characteristics and Pathogenicity of a Novel Porcine Epidemic Diarrhea Virus with a Naturally Occurring Truncated ORF3 Gene"

_viruses, 2022, doi:10.3390/v14030487_

Round 1

Reviewer 1 Report

Dear Authors,

Congratulations for this huge work around the PEDV virus it represents an interesting value added in the field.

On the other hand, I think that the presentation should be improved and that it should be easier to read this big work. You will find my remarks and proposals in the points below. I have proposed a major revision because I believe that some parts need to be reorganized and especially the figures improved. I do not challenge the overall quality of the work but rather the presentation.

Major revision:

  1. I find that there are a lot of different anaylses that have been realized, maybe a flowchart could facilitate the reading and the understanding of the different steps performed.
  2.  all phylogenetic trees are not readable and can be improved on the shape and size of the tree. the branches are much too small compared to the colored circles. The trees really need to be made better
  3. Most of the discussion is not referenced enough, this needs to be improved
  4. The discussion is long, I think it is necessary to be more synthetic on many points so that it is simpler to read and not to describe at this level the results

Minor revision:

Line 83-84 Add the number of ethical agreement

Line 96-97 It might be better to add the primers M and ORF3 in Table S1 so that this table summarizes all the primers used

160-164 I don't understand why we have CT values to evaluate the level of clinical symptoms ? Can you explain this point better in the paragraph ?

Reviewer 2 Report

Summary

Porcine Epidemic Diarrhea Virus, the causative agent of epidemic swine diarrhea, is responsible for numerous infections in pigs worldwide. Despite various vaccination programs, PED still causes tremendous economic problems in the swine industry, indicating the need for better prevention programs. Therefore, information of PEDV outbreaks should be monitored and investigated.

In the present study “Genetic Characteristics and pathogenicity of novel porcine epidemic diarrhea virus with a naturally occurring truncated ORF3 gene” the authors characterize four different PEDV strains, which were circulating in pig farms in central china during 2012 to 2021. Genomic analysis based on the complete genome and S and ORF3 genes revealed that the newly characterized PEDV strains belong to the variant G2a subgroup, in which other strains circulating in China are also classified. In addition, all strains examined exhibited amino acid changes in the S protein, particularly in the major neutralizing epitopes, compared with the vaccine strain CV777. In addition, the author demonstrated that the newly described strain HN2021, which was successfully isolated and propagated in cell culture, contains a naturally occurring truncated ORF3, resulting in a truncated protein with only 68 amino acids. Nevertheless, animal studies in piglets showed that the truncated HN2021 strain is a virulent strain that causes severe diarrhea and high mortality, which could make it a promising vaccine candidate. Overall, the current study is well conducted and the experiments were well executed. However, there are some points that should be revised to improve the quality of the article.

Dear authors,

Thank you for your interesting manuscript on characterization of different strains of Porcine Epidemic Diarrhea Virus. In my opinion, the manuscript is well developed, but there are some aspects that should be revised to improve the submitted article.

Major points:

Line 251: The recombination event of which strain in shown in figure 3d? First, the text says that a recombination event was found in CH-HNYY-2018 but the next sentence says that figure 3d shows strain HN2021. Please clarify this.

Line 388: It might be confusing for readers that line 214 states that the ORF3 gene of CV777 is "complete", while line 387 mentions that 49 nt are deleted in the classical vaccine strains including CV777. Please explain these differences. 

Overall, all the images are of very poor quality and have to be improved.

-When printing the document, it was not possible to read the results for Table 1, and even on the computer screen the font is too blurry to read the rows correctly.

- All images of figure 3 are too small. It is not possible to see the lines of the phylogenetic tree. In figure 3d it is not possible to read the writing that is on the image.

- Also, I would like to ask you to recreate the image of the plaque purification assay in Figure 4d, as the image is also very blurry and the plaques are not easy to see.

Last but not least, in my opinion, the complete text needs a comprehensive revision in terms of grammar and style.

Round 2

Reviewer 1 Report

Dear Authors,

Thank you for your answers. All figures and tables are still of very bad quality it is not possible to read them. Also the phylogenetic trees could be improved because the colored part takes too much space compared to the tree itself and the names of sequences etc. are unreadable. I think it is necessary to change this before publication.

Kind regards.

Author Response

Date: 2-19-2022; RE: viruses-1593729

Dear reviewer,

We would like to thank Viruses for giving us the opportunity to revise our manuscript. We appreciated the concerns and suggestions, and we have revised our manuscript accordingly. All the figures and tables were improved again to be readable. However, Table 1 is so large that it may not be clear when placed on the manuscript page, so we adjust the font size in the table to 12 pt. If there is still ambiguity, we would like to put table 1 in the supplementary materials of the manuscript.

Thanks for all the help.

Best wishes,

Prof. Hong-Ying Chen

Reviewer 2 Report

Dear Authors,

Thank you for responding to my main points. Unfortunately, the quality of almost all figures and tables shown is still very poor. Especially the presentation of the phylogenetic analysis is so small that all information is lost. The quality needs to be improved before publication. 

Author Response

(The authors gave the same response as above.)
